An artificial intelligence-based classifier for musical emotion expression in media education

Lian Jue lianj94726@163.com
Faculty of Creative Arts, Music Department, University of Malaya , Kuala Lumpur , Malaysia
Asif Muhammad
Electronic publication date: 2023 Jul 14
Publication date: 2023
Volume: 9
Electronic Location ID: e1472
Received 2023 Apr 5; Accepted 2023 Jun 8
Copyright: © 2023 Lian
Copyright year: 2023
Copyright holder: Lian
License: This is an open access article distributed under the terms of the Creative Commons Attribution License, which permits unrestricted use, distribution, reproduction and adaptation in any medium and for any purpose provided that it is properly attributed. For attribution, the original author(s), title, publication source (PeerJ Computer Science) and either DOI or URL of the article must be cited.
License URL: https://creativecommons.org/licenses/by/4.0/

Keywords: Emotion expression, RBF classifier, IoT, Audio features, New media education

Funding: The author received no funding for this work.

==============================
Music can serve as a potent tool for conveying emotions and regulating learners’ moods, while the systematic application of emotional assessment can help to improve teaching efficiency. However, existing music emotion analysis methods based on Artificial Intelligence (AI) rely primarily on pre-marked content, such as lyrics and fail to adequately account for music signals’ perception, transmission, and recognition. To address this limitation, this study first employs sound-level segmentation, data frame processing, and threshold determination to enable intelligent segmentation and recognition of notes. Next, based on the extracted audio features, a Radial Basis Function (RBF) model is utilized to construct a music emotion classifier. Finally, correlation feedback was used to label the classification results further and train the classifier. The study compares the music emotion classification method commonly used in Chinese music education with the Hevner emotion model. It identifies four emotion categories: Quiet, Happy, Sad, and Excited, to classify performers’ emotions. The testing results demonstrate that audio feature recognition time is a mere 0.004 min, with an accuracy rate of over 95%. Furthermore, classifying performers’ emotions based on audio features is consistent with conventional human cognition.

Introduction

With the advancement of Internet technology, learners can use mobile phones, computers and other electronic devices at home to log on to various learning platforms. Online learning places teachers and students in the same space, where teachers can display the teaching content on the network platform to teach audio-visual synchronization. The progress of Internet technology has solved the technical problems of online learning. Efficient online music education requires students to practice and experience music skills repeatedly. Applying the Internet of Things (IoT) technology to music education can enable students to strengthen their knowledge and memory learned in class (Zhao, 2022). Among them, music feature recognition is one of the difficulties in realizing online music teaching, an art developed based on speech recognition. It obtains music content through audio signals and features such as music form and emotion (Kai, 2021). Currently, the music recognition system cannot be widely used because the overall system framework design is not conducive to improving the performance information. IoT technology enables intelligent music signal collection, processing, and analysis via real-time information transmission between networks (Baratè et al., 2019). For example, in music teaching, IoT can record and store teaching videos and share teaching audio and video resources so that more people can watch and learn repeatedly. Applying the IoT technology to music education is a complementary advantage between the Internet and music teaching, where teachers can spread more teaching resources and videos. Besides, learners can download more music education videos from the Internet and also download some online music for analysis to learn the advantages and characteristics of online songs in creation, which provides students with a more diversified and multifaceted learning environment and better improves the accumulation of their music knowledge (Liu & Yang, 2021; Li, 2019).

Music classes can allow students to express themselves; for example, in the knowledge teaching of “Reforming Rivers and Mountains to Wait to born later”, teachers can let students learn folk music and stimulate the emotional experience of folk music. In the study of “Homesick”, students can feel the creator’s feeling of homesickness. However, song implication and emotional expression are difficult to spread effectively in online learning. The essence of music lies in its possible emotional expression (Jian, Lei & Qi, 2019). Currently, the analysis of data features found in music texts and audio signals is used more to classify and recognize musical emotion than other methods (Kun & Lixin, 2018; Medina, Beltrán & Baldassarri, 2020). The recognition model of music feature signals is built by finding features like timbre, rhythm, and rhythm through appropriate learning methods. For instance, Bagofwords (Chao, 2018) may classify emotions based on song lyrics according to various musical modes, and the proper expression programming method is employed to do so regarding music rhythm (Bulagang et al., 2020).

Online music instruction is poised to offer a captivating alternative to traditional classroom pedagogy, affording students unprecedented opportunities to acquire musical skills irrespective of temporal, spatial, and financial constraints. Nonetheless, a lingering challenge faced by music educators lies in their ability to promptly discern fluctuations in students’ emotional states during song interpretation, underscoring the need to explore Internet-based technologies tailored to online music instruction. By doing so, we can effectively bolster the quality of music literacy, artistic assessment, and overall teaching standards, ultimately driving the evolution of a new and personalized online instructional model.

This article firstly designs a music feature recognition system based on IoT technology, which realizes the rapid, accurate and intelligent segmentation and recognition of notes. Then, the identified music signals are classified by RBF neural network. Generally speaking, the Internet of Things is an extension and expansion of the network based on the Internet. It combines various information-sensing devices with the Internet to form a vast network which can realize the interconnection of people, machines and things at any time and anywhere. In addition, RBF (Radial Basis Function) is a single hidden layer feedforward neural network that uses the radial basis function as the hidden layer neuron activation function. In contrast, the output layer is a linear combination of the output of the hidden layer neuron. Its main contributions are as follows:

Segmentation is carried out based on the changing trend of physical characteristics of musical notes, where the first segmentation is realized by the start-stop time and pause of notes and the second segmentation is completed according to the voice segment, noise segment and vocal segment of music, thus completing the division of a frequency spectrum into independent note tones.

The proposed method can fuse multi-source information and improve the accuracy of multi-classification. When classifying music emotions, the emotional tags marked by user groups represent the user’s subjective experience, which means the user’s emotional perception of music after deduction. The emotional recognition results are closer to the user’s personal feelings.

This article integrates the human-computer interaction technology of the IoT and artificial intelligence algorithm, which makes different people form an excellent teaching loop in online music education and provides a new idea for realizing the interactive teaching method of music and intelligent mobile terminals.

Related works

Music feature recognition

Currently, the most common method of multimodal music feature recognition is the combination of lyrics and audio. Huertas-García et al. (2022) adopted the multimodal music emotion classification method combining audio and lyrics and improved the traditional fusion sub-task merging method LFSM (late fusion sub-task merging). Existing decision-level fusion methods model multimodal data separately and then linearly weight the obtained decisions to generate the final result (Zhang et al., 2020). Despite the effectiveness of linear weighted decision-making level fusion in enabling multiple categories of a single pattern to share the same weight and be accurately computed through weighted averaging, this method cannot, unfortunately, assign varied weights to distinct categories. Pandeya & Lee (2021) introduced a neural network to improve the decision-level fusion method. Compared with the traditional linear weighted decision-level multi-mode fusion method, the enhanced multi-mode fusion method has a 6.4% improvement effect, but there is still much room for improvement. In the traditional methods, LORFA spectral feature extraction and high-order spectral feature extraction are mainly used for music signal feature extraction, which combines adaptive signal separation and blind source filtering for music recognition (Li, Yu & Hu, 2021; Wang & Peng, 2020), which has a certain recognition level, but with the increase of synthetic components in music signals, the recognition accuracy is not good. Zhang, Meng & Li (2016) extracted the signal diagram of song audio, converted the signal diagram into the frequency spectrum diagram, and realized the separation and mixing of music through image recognition and analysis. It still adopts the image recognition method but does not realize the actual multimodal fusion.

Among the existing methods, the machine learning method using audio bottom features is blind and unknown. The cost of comparing the combination of audio bottom features in a wide range is too high. Meanwhile, the traditional audio feature recognition method cannot solve the problem of speech recognition in a noisy environment in online education. IoT technology realizes the real-time transmission of music signals through wireless networks, which can quickly collect, process and analyze them.

Musical, emotional expression based on deep learning

In recent years, the accuracy of using deep learning methods to identify musical emotions has been dramatically improved (Jian et al., 2022). Most of the deep learning-music-emotion-recognition methods are based on the neural network model, and the design of its network model affects the recognition accuracy, which is suitable for dealing with large sample data. The task of analyzing emotions in computer-generated music necessitates a process of mapping high-dimensional spaces to lower dimensions. However, owing to the inherent ambiguity of emotional expression, relying solely on underlying features for computer-based classification can prove inadequate in accurately capturing the nuances of musical emotion. In response, relevance feedback technology is employed to refine and enhance the classifier, thereby optimizing its efficacy. Considering the impact of local key information on music emotion, Koh & Dubnov (2021) aggregated the high-dimensional spectrogram features by a deep audio embedding method using L3-Net and VGGish convolutional neural networks. This was done to determine the music emotion. However, a convolution neural network does not consider the timing of musical emotion, so only using a convolution neural network or cyclic neural network cannot solve the problem of musical emotion recognition. Xia, Chenxi & Jiangfeng (2019) proposed a deep learning model for this, combining two-dimensional CNN and RNN to analyse spectrogram features to identify musical emotion. In the extraction process of music signals, MFCC is usually combined, for example, Yuqing et al. (2019) fully extracted the emotion features in the time-frequency domain of the spectrogram. They proposed a method combining parameter migration and convolution recurrent neural networks to recognise speech emotion. Hizlisoy, Yildirim & Tufekci (2020) based on the standard acoustic statistical features, combined with the Mel cepstrum coefficient (MFCC) spectrogram and Mel filter group energy spectrogram features, to identify musical emotion. Li et al. (2020) fused MFCC and MIDI audio features, then fused twice with lyrics text features. The effect of the first feature fusion was better than that of a single feature, and the classification accuracy of the second feature fusion was further improved. The above research on multimodal music emotion classification has turned audio features and text features into numerical values, and some scholars used spectrograms to represent audio features.

From the above analysis, it can also be seen that musical emotion analysis is inseparable from feature extraction and quantification. Especially in online education, the music features of the music works performed by students are not as good as those published. Meanwhile, when the users guide the search to express teaching emotion, the feedback results will develop in a direction favourable to users’ requirements. Thus, in light of these limitations, collaboration between humans and machines becomes essential to compensate for the shortcomings of computer-based understanding. Integration of Internet of Things (IoT) technology with deep learning represents a promising avenue for bridging the gap between users’ high-level semantic concepts and the underlying characteristics of music, thereby significantly enhancing the accuracy of emotional interpretation and ultimately improving the effectiveness of online music instruction.

Intelligent music feature recognition

Overall structure

The overall structure block diagram of the proposed system is shown in Fig. 1. The physical layer mainly includes music signal acquisition and processing modules (Ye & Mohamadian, 2016). The former uses sound sensor acquisition systems arranged in various positions to find the required music signal. It then sends the music signal to the music signal processing module, which uses the DSP processor to process the music signal. While the latter sends the information gathered and handled by the actual detecting layer to the framework application layer through remote organisation correspondence transmission.

Figure 1 System architecture.

The module includes a music acquisition sub-module and a voice coding sub-module. The music acquisition sub-module comprises sound sensors installed in different positions responsible for acquiring the original music signals.

Audio feature segmentation

In the realm of human-computer interaction for the Internet of Things, the perception layer can effectively capture online audio recordings of students’ music interpretation. This audio can then be transmitted via network communication to enable the division of music melody based on the start and end times of individual notes. This approach is advantageous in improving the efficiency and accuracy of music melody classification. Once an error occurs, the accuracy of the segmentation result will be reduced. In this article, the note segmentation recognition method of audio feature technology is used to cut the note, and the specific implementation process is shown in Fig. 2.

Figure 2 Audio feature segmentation process.

(1) Firstly, the selected musical notes are segmented for the first time according to the sound level, so as to reduce the workflow for the cutting of the later notes.

(2) The first segmented notes are subjected to data frame framing, spectrum mapping and feature extraction according to the 12-average mapping principle.

(3) Finally, the extracted note features are determined by the step length of note cutting and the threshold between each note. After the key data of note cutting is determined, the notes of each feature are cut twice, and the intelligent music note segmentation and recognition operation is completed.

For musical notes, frequency, amplitude, intensity, pitch, duration and temperament are the main physical characteristics of musical notes. Each physical characteristic shows different features of musical notes, and these essential characteristics play a significant role in composing chords, rhythms and melodies of middle and high levels of music (Li et al., 2020). The relationship between musical notes is shown in Fig. 3.

Figure 3 Relationship between notes.

The step size of notes is determined according to the tone level, melody and sound intensity of melody notes. Firstly, the energy value of each frame in the whole speech signal should be taken as the research object and considered as a whole. Calculate the maximum and minimum values of short-time energy of all note frames, randomly divided into z equal parts, where z is randomly determined according to the melody length. According to the length of the melody. The step of a note is the quotient of the absolute value of the difference between the maximum and minimum value of the note data frame with z. The calculation formula is shown in Eq. (1):

(1) W=max−minz.

When satisfied |zi+1−zi|<1, the returned threshold is B= min+(i+1)∗ Step size; otherwise, i=i+1. Calculate the step length of repeated notes until the appropriate note-cutting threshold is calculated. The step length corresponding to this threshold is the cutting step length of actual notes. Once the energy of a note is too small, it will be ignored in the intelligent segmentation of notes. At this time, if the recognition data of a note is slightly cut in the recognition result, it can be cut again to improve the intelligent segmentation result of notes and improve the accuracy of note segmentation.

The music feature analysis adopts a dynamic time-warping (DTW) algorithm (Chen et al., 2016). It identifies the music signal features by comparing the Euclidean distance between the music feature test and reference templates.

Assume that the grid points that the path passes through, in turn, are (n1,m1),⋯, (nI,I),(nN,mM), which can be obtained according to the endpoint constraints. (N1,m1)= (1,1),(nN,mM)=(N,M).

The final cumulative distance of the path is as follows:

(2) L[(Ii,mi)]=l[T(I,R(mi))]+L[(ni−1,mi−1)].

The minimum cumulative distance is the result of music signal feature recognition.

Musical, emotional expression

Emotional classification in music education in China

Music emotion classification is the first step of emotion analysis. The Thayer and Hevner emotion models are commonly used in music (Kim & Belkin, 2002). China’s music education has a certain national particularity. Based on the two classification methods, this paper conducts a quantitative expression of different word lists according to the structure of Hevner by conducting an online survey and interview with music experts to meet Chinese “student” learning habits. After modification and deletion, four emotional categories were finally confirmed, namely Quiet, Happy, Sad and Excited, as shown in Fig. 4.

Figure 4 Corresponding relationship between music emotion classification in Chinese music education and Hevner emotion model.

Parameterised emotional characteristics

Based on the note features extracted from “Audio feature segmentation”, music emotion analysis needs to consider the basic music information, rhythm changes and structural forms and divide the music features into note features and high-level features. Pitch, duration and intensity, as the essential components of music, are acoustic clues for music emotion cognition. The pitch change of music is represented by the mean square deviation of pitch, as shown in Eq. (3).

(3) varP=1n∑i=1n[Pi−∑i=1nPin]

where, Pi represents the pitch of the i-th note; n is the number of notes. The range is used to indicate the pitch span of a music and define the range, as shown in Eq. (4).

(4) Range=Max(P1,P2,⋯Pn)−Min(P1,P2,⋯Pn)

where, P1,P2,⋯Pn Indicates the Pitch value of the note.

Duration is the author’s duration, and the expression method is shown in Eq. (5).

(5) Duration=EndTime−StartTime.

Dynamics is an important means for music to express emotions; different musical dynamics can cause different emotional experiences for listeners. The average strength and strength variation characteristics can be characterised by the average dynamics represented by the Eq. (6).

(6) Dyn=1m∑i=1m[Ii−∑i=1MIin]

where n and m indicate the number of musical notes, respectively. The degree of dynamics variation is calculated by Eq. (7).

(7) varDyn=∑i=1N−1Di+1−DiN⋅Dyn

where, Di represents the intensity of the i-th musical note; N is the number of notes. Equation (7) describes the change of dynamics using the music bar, excluding the influence brought by the weight change in the beat.

Melody reflects the organisational form of music in time and space, and it expresses the change of pitch trend. For this reason, melody direction is expressed, such as Eq. (8).

(8) Mel=∑i=1n−1(Pi+1−Pi⋅Di)D−Dn

where, Di represents the i-th sound length; D stands for sum of length.

Rhythm is a phenomenon of alternating regular intensity in music. Different festivals give music different tension. According to the tension change of rhythmic movement, the density of pronunciation points directly reflects its basic state. The greater the density, the more muscular the tension. Sign the change of rhythm density, as shown in Eq. (9).

(9) Mutation=Max(|BarCapacity(i)−BarCapacity(i−1)|Max(BarCapacity(i)))

where BarCapacity(i) represents the energy value in i-th bar, MaxBarCapacity(i) indicates the maximum energy of the bar.

Classification of music emotions based on RBF

(1) After extracting the music features that can express emotion, the music emotion classifier is constructed. In this algorithm, the three-layer radial basis function neural network is used as the intelligent classifier, which has the best approximation and global approximation and is superior to the traditional BP network in function fitting. The algorithm flow is as follows: firstly, a record is randomly selected from the library, and similar records in the library are searched; In the results found by users, matching records are marked and fed back to the system, and the system marks them in the database; The system uses these marked records to train the classifier until the searched false emotion fragments are within a certain error range. In the process of correctly constructing the classifier, the emotional annotation of music in the library is realised. The construction algorithm of the classifier is as follo(1). The feature vector f1f2⋯fn was extracted from n music pieces in the library, where fi={fi1,fi2,⋯,fim}; Because of the different physical meanings of the m components in fi, this article uses the Gaussian function to normalise their features, as shown in Eq. (10).

(10) fij=fij−mj3σj,1≤j≤m

where: mj is the mean value; σj is standard deviation, m = 15.

(2) A fragment ai from n music fragments are randomly selected, and its characteristic is the vector fi. Calculate the distance between k and other music fragments in the library according to Eq. (11), and return l results with smaller distances to the learners.

(11) d=∑j=1m(fij−fkj)2

(3) The trained classifier is used to classify the last marked fragments in the library and return them to the user. The user judges these results, draws them in the library if they are fragments with similar emotions, and jumps to Step 4 for execution. Otherwise, return to Step 2.

(4) Return to Step 1, and circularly execute Step 1 to Step 4 until all student inputs are classified.

Experiment and analysis

Dataset

This test adopts AMG1 608 data set (Chen et al., 2015) which contains 1,608 30 s music clip, and 665 subjects annotate the clips. The data set includes annotations of 46 themes, and each theme is annotated with more than 150 music pieces. The speech coding submodule plays a critical role in achieving high-fidelity, lossless compression of the original music signal. This entails converting the music signal into data that can be readily transmitted, before relaying it to the music signal processing module for further analysis. The joint representation of each song in the data set is calculated, and finally each audio segment is mapped to the vector form, forming the representational corpus of the seed audio. By searching this corpus, the joint representation of any kind of audio can be obtained without repeated calculation, which provides support for subsequent research.

Music feature recognition results

In order to ensure the rationality and scientificity of this experiment, this article compares the methods used in literature (Calvo-Zaragoza, Hajič & Pacha, 2020; Wen et al., 2015) and divides and recognises the same musical melody according to three methods. Record the notes, and each group of experiments is conducted three times. Finally, balance three "time” processing time and note segmentation’s effect as the final experimental result.

From the analysis of Fig. 5, different methods have different recognition times for note segmentation. When the system identification amount is 2 GB, the compared “method” identification time is 0.037 and 0.030 min, while the identification time of this method is only 0.004 min. The recognition time of this method is much lower than that of the other two ways. The recognition accuracy of different methods is shown in Fig. 6. The designed method has an accuracy rate of nearly 100%.

Figure 5 Identify time results.

Figure 6 Recognition accuracy results.

The music signals of two different scenes in an online learning segment are shown in Fig. 7.

Figure 7 Music signal acquisition results.

From Fig. 7, The author’s collection of the music signal has a smooth curve with no burr or signal interruption, indicating stability, and the acquired music signal has good sound quality.

Emotional classification results

In order to verify the effect of the classification model, a comparative experiment was conducted between the proposed classification model and the BP neural network-music-emotion classification model used in Chun, Song & Yang (2014).

In Fig. 8, the recognition results of two different algorithms are compared. Because IoT technology is used to segment and recognise musical notes intelligently, the multi-classification algorithm in this article has the largest number of samples of four emotions accurately recognised, which is superior to the BP neural network and has higher generalisation and stability.

Figure 8 Accuracy of emotion classification.

The confusion matrix of models under different emotions is shown in Fig. 9. The model has the highest recognition accuracy of Sad, reaching 90%, and the lowest recognition accuracy of happy, less than 80%, which indicates that “student” sad emotional expressions will be prominent and can be recognised more accurately. At the same time, happy songs have many complex deductive forms, such as rap songs. The length of rap lyrics is generally larger than other types of songs. Rap lyrics’ content is relatively diverse; therefore, machine recognition will have some difficulty with them, leading to a worse classification effect than the other three categories. By analysing the wrongly classified songs in happiness, it is found that most of them are Chinese love songs describing love, while sad songs are almost all composed of Chinese love songs, while quiet songs are composed of less than half of Chinese love songs. The only exciting songs are nearly all foreign songs. Due to linguistic and cultural barriers, although many love songs are present in exciting foreign songs, there is a small amount of confusion regarding happiness and excitement. Still, there is much confusion regarding sad and quiet lyrics. Generally speaking, classifying the “performer’s” emotions based on audio features can produce results that align with people’s conventional cognition.

Figure 9 Confusion matrix of emotion classification.

Discussion

It is evident from the initial experimental findings that the proposed method realises the rapid segmentation of musical note features because the technique in this article can segment musical notes according to the changing trend of their physical features. For a musical melody, the hierarchical filtering segmentation method is adopted to complete the operation, which effectively extracts the features of musical notes, shortens the recognition time, and provides a foundation for “student” emotional expression in online teaching. Music emotion recognition is the basis of “student” emotion recognition in online learning. With the continuous expansion of the online teaching scale and the daily explosive growth of massive data, online education platform needs to classify music in time. It can accurately recommend new music to users in need. The proposed musical emotional expression can obtain the emotional tendency of many songs. Technically, it can fuse multi-source information and improve the accuracy of multi-classification. In this article, when classifying music emotion, the emotional tags marked by user groups represent the users’ subjective experience, which is the users’ emotional perception of music after listening to music. The emotional recognition results are closer to the users’ personal feelings.

Although the current interactive music platform Wolfie combines artificial intelligence technology with a music database to support the “student” musical performances (Shang, 2019), the learning efficiency is significantly increased by using multimedia assistance, including the instructor model and the learner model, to realise the integrated interactive teaching method of intelligent mobile terminals and reduce the amount of human involvement (Zhao, 2022). The online teaching scheme proposed in this article focuses on timely feedback on “learner’s” emotional state to learners, forming a good teaching loop, as shown in Fig. 10. Music learners present their learning results to the teachers through the human-machine interaction mode in the online learning platform. The educators further develop the new showing mode through their accomplishments and develop the “student” learning mindfulness to learn in view of negative feedback, subsequently shaping an ideal intelligent shut circle.

Figure 10 Interactive online music learning architecture.

Conclusion

In this article, a music feature recognition system based on IoT technology is designed. The feature extraction analysis is carried out by analysing the physical characteristics of music notes, and the music signal feature recognition is realised, where musical emotional expression is then identified according to the results. Regarding feature extraction results, the time of audio feature recognition using this method is only 0.004 min, and the accuracy rate is higher than 95%. In addition, different languages have certain influences on learner pronunciation, which leads to a small degree of confusion between happiness and excitement, but a large degree of confusion with sad and quiet lyrics, In practical testing, it was observed that happiness was less likely to be confused with excitement but was more prone to be mistaken for sad and melancholic lyrics, which could potentially impede a teacher’s ability to effectively manage student emotions. To address this issue, future research will employ a fine-grained emotion analysis model to investigate the intersection between positive emotions, and to further refine “student” emotional classification for improved accuracy.

Supplemental Information

Supplemental Information 1 Code.

Click here for additional data file.

We thank the anonymous reviewers whose comments and suggestions helped to improve the manuscript.

Additional Information and Declarations

Competing Interests

Author Contributions

Data Availability

The author declares that they have no competing interests.

Jue Lian conceived and designed the experiments, performed the experiments, analyzed the data, performed the computation work, prepared figures and/or tables, authored or reviewed drafts of the article, and approved the final draft.

The following information was supplied regarding data availability:

The code is available in the Supplemental Files.

The data was obtained from Kaggle: https://www.kaggle.com/datasets/snapcrack/all-the-news.

The AMG1608 Dataset is available at GitHub: https://github.com/loichan-tw/AMG1608_release.

The AMG1608 Dataset came from: The AMG1608 dataset for music emotion recognition. In 2015 IEEE international conference on acoustics, speech and signal processing (ICASSP), pp. 693–697. Piscataway: IEEE, 2015.

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
