# Peer review of "An artificial intelligence-based classifier for musical emotion expression in media education"

_PeerJ Computer Science, doi:10.7717/peerj-cs.1472_

## Round 0.1 · original submission · Major Revisions

As suggested by the experts, a couple of major changes are required. Therefore, we advise you to carefully revise the article in light of the comments and resubmit for further consideration.

·

Basic reporting

The author propses the intelligent segmentation and recognition of musical notes by first segmentation, data frame segmentation and threshold determination. Then, according to the extracted audio features, the music emotion classifier is built by using RBF. Finally, the classification results are marked by correlation feedback and the classifier is further trained. By comparing the classification of music emotions commonly used in Chinese music education with Hevner's emotion model, four categories of Quiet, Happy, Sad and Excited are selected to classify the emotions of performers. The test results show that the classification of emotions based on audio characteristics can produce results consistent with people's conventional cognition. This paper has some innovation, but it needs to be modified.
1. However, there are some questions to be clarified for the exact contributions and the description of the methods achieved in this work as follows.
2. In general, there is a lack of explanation of replicates and statistical methods used in the study.
3. Furthermore, an explanation of why the authors did these various experiments should be provided.
4. In order to help readers better understand the topic of the article, the author gives some examples, but they are still very difficult to understand. The addition of some quotes helps to change this situation;
5. In the literature review, the author did not put forward his own unique insights, which need to be combined with the main contribution of the research;
6. The author identifies the desired music signals through a sound sensor acquisition system placed at different locations. The ethical issues of this work should be considered;
7. The experimental part lacks the comparison of the application of the Internet of Things technology. I haven't seen more data description of the actual test;
8. The limitations of the research and the future development direction need further elaboration and analysis.

Experimental design

no comment

Validity of the findings

no comment

Additional comments

I have no additional comments

Reviewer 2 ·

Basic reporting

The interactive online music teaching system provides better technical support and learning ideas for music learners. The teaching model based on artificial intelligence can reasonably apply emotional evaluation to improve the teaching efficiency of music learners. In this paper, a music feature recognition system based on the Internet of Things technology is designed. By analyzing the physical characteristics of music notes, feature extraction and analysis are carried out. In the music feature analysis module, dynamic time normalization algorithm is adopted to obtain the maximum similarity between the test template and the reference template to realize the feature recognition of music signals. And according to the recognition results, the corresponding musical feature content of musical form and musical emotion is identified. But there are still parts that need to be revised as follows:

1. There are still some language expression problems in this article, which need to be further modified. Please check the full text and deal with it.
2. The form of reference in the article is very strange, just keep one paradigm;
3. The background analysis of the abstract is not enough. The contents of the main issues should be elaborated.
4. Also, there are few explanations of the rationale for the study design.
5. Most of the documents in Section 2.1 are old, please replace some descriptions of recent years;
6. If the Internet of Things technology is only used for audio capture, it won't be a major innovation;
7. Why is "Hevner affective ring model" used to construct affective indicators?
8. The conclusion part needs to adjust the language expression, and the elaboration of this part is too lengthy. The author needs to simplify this part.

Experimental design

no comment

Validity of the findings

no comment

Additional comments

no comment

---

## Round 0.2 · accepted · Accept

Based on the outcome of the peer review process and the input of experts, your paper has been accepted for publication. Congratulations!

·

Basic reporting

The authors have made the required comments in the revised paper. It looks great.

Experimental design

The authors have made the required comments in the revised paper. It looks great.

Validity of the findings

The authors have made the required comments in the revised paper. It looks great.

Additional comments

I have no additional comments.

Reviewer 2 ·

Basic reporting

All my concerns have been addressed!

Experimental design

no comments

Validity of the findings

no comments